# Possibilities of Using De Novo Design for Generating Diverse Functional Food Enzymes

**DOI:** 10.3390/ijms24043827

**Published:** 2023-02-14

**Authors:** Xinglong Wang, Kangjie Xu, Yameng Tan, Song Liu, Jingwen Zhou

**Affiliations:** 1Engineering Research Center of Ministry of Education on Food Synthetic Biotechnology, School of Biotechnology, Jiangnan University, Wuxi 214122, China; 2Science Center for Future Foods, Jiangnan University, Wuxi 214122, China; 3Jiangsu Province Engineering Research Center of Food Synthetic Biotechnology, Jiangnan University, Wuxi 214122, China

**Keywords:** food enzyme engineering, artificial intelligence, enzyme engineering, de novo design

## Abstract

Food enzymes have an important role in the improvement of certain food characteristics, such as texture improvement, elimination of toxins and allergens, production of carbohydrates, enhancing flavor/appearance characteristics. Recently, along with the development of artificial meats, food enzymes have been employed to achieve more diverse functions, especially in converting non-edible biomass to delicious foods. Reported food enzyme modifications for specific applications have highlighted the significance of enzyme engineering. However, using direct evolution or rational design showed inherent limitations due to the mutation rates, which made it difficult to satisfy the stability or specific activity needs for certain applications. Generating functional enzymes using de novo design, which highly assembles naturally existing enzymes, provides potential solutions for screening desired enzymes. Here, we describe the functions and applications of food enzymes to introduce the need for food enzymes engineering. To illustrate the possibilities of using de novo design for generating diverse functional proteins, we reviewed protein modelling and de novo design methods and their implementations. The future directions for adding structural data for de novo design model training, acquiring diversified training data, and investigating the relationship between enzyme–substrate binding and activity were highlighted as challenges to overcome for the de novo design of food enzymes.

## 1. Introduction

Proteins are widely distributed in all living organisms, and are natural biocatalysts that participate in biological reactions [1,2,3,4]. Proteins are widely applied in various industries. They are used for generating food appearance and flavor [5], participating in (bio)-material processing [6], or used as drugs [7] or bio-materials [8]. Food enzymes are a group of proteins that are emerging as additives used for food processing [9]. In recent years, studies have shown that food enzymes not only play a role in traditional food industrial processing, such as in baking, dairy, bean and meat products [10], but have also been advantageous to novel food products processing, such as artificial meat [11]. In addition, several food enzymes show capabilities for degrading allergens [12] or bitter peptides [13], providing food processing convenience. Since the sophisticated performance of food enzymes was approved, the US Food and Drug Administration (FDA) and European Food Safety Authority (EFSA) have issued several statements to standardize the uses of food enzymes, in order to ensure food safety [14]. With the permission for use, the applications of enzymes in food processing continuously increased over the past few years; the reported market size for enzymes (mainly food enzymes) exceeded 17 billion in 2020 [15,16]. However, not all food enzymes can meet their industrial application needs, partly due to their insufficient stability or low activity, and partly due to adverse processing conditions [17].

Enzyme engineering aims to resolve weak enzyme stability and low activity issues. The enzyme engineering technique can be mainly divided into direct-evolution, rational design, and semi-rational design [18]. Although enzyme engineering still massively relies on high-throughput screening supported by direct evolution and semi-rational design, reports on rational design-based enzyme modification are displaying an increasing trend [19]. One reason is the high labor and experimental cost of high-throughput screening, and it is always hard to find a screening method [20]. Another reason is that the rise in accurate protein modeling and energy computing techniques in recent years has dramatically reduced the difficulty of rational design [21,22]. Therefore, a growing number of researchers are adopting rational design for enzyme evolution. Developing protein design methods has served the need for screening functionalized therapeutic proteins or high-efficiency enzymes [23,24], which aim at satisfying the grand market of commercialized therapeutic proteins [25]. Rational design methods successfully assisted protein evolution [21,26,27,28]. However, it is difficult to substantially improve a certain function of a given protein by conducting single or several point mutations, or by bringing in highly diversified changes to protein functions.

A breakthrough in the field of rational design is de novo design. De novo design learns features from existing sequences and structures, which can be used to create diversified novel proteins to reinforce the functionality of natural proteins [29,30]. The progression of AI technology and the development of bioinformatics have pushed the rapid development of de novo design techniques to increase the number of proteins that were designed from scratch displaying diverse functions [31,32,33,34]. Advances in protein de novo design have provided opportunities for the direct evolution of enzymes. The implementation of de novo design for generating functional proteins, protein binders or industrial enzymes has achieved great success [31,32,33,34,35]. In this review, by describing the function and challenges of food enzymes, the potential and opportunities of using de novo design techniques for functional food enzymes engineering were highlighted. The AI-based protein modeling and de novo design tools, as well as their utility for protein engineering, were reviewed to gain insight into the potential of using these tools for food enzymes engineering.

## 2. Diverse Functions of Food Enzymes

Food enzymes are widely used in various food processing [19], and are mostly derived from microorganisms. According to enzyme classification, commercial food enzymes can be classified into oxidoreductases, transferases, hydrolases, isomerases and ligases, while proteases take the principal portion (Table 1). In practical uses, food enzymes are mainly used to improve food taste and appearance, or to convert sugar-related reactions [36,37,38] (Figure 1). For example, in meat product processing, proteases (such as papain) and transglutaminases are used for meat tenderization [39] and improving texture [36], respectively. In dairy products processing, proteases such as lactase can be used for hydrolyzing lactose into galactose and glucose, in order to assist human absorbance [40]; lipase can assist in oil hydrolysis [41]; and esterase can improve the flavor by hydrolyzing esters into acids and alcohol [42]. For winemaking, pectinase can hydrolyze pectin to improve the flavor and color of wine [43], and glucose oxidase can oxidize glucose into gluconic acid and generate hydrogen peroxide, which can improve product quality and enhance the storage period during beer production [44].

The application of food enzymes in food processing has gradually been recognized with the growing attempts to use food enzymes. Several food enzymes, such as transglutaminases, laccases and lactases, can be used in processing meat, dairy, and bean products [36,40,45]. Traditional food enzymes such as peroxidase, transglutaminases and laccases are well suited for improving meat texture and flavor for novel products such as artificial meat. Recent studies showed that some food and several potential food enzymes (not commercialized) are essential in desensitization and flavoring. There has been research on using aldehyde dehydrogenase [56] and aldehyde oxidase [57] to remove the beany flavor and foul smell of soybean-based artificial meat. Studies on enzymatic degradation of soybean protein allergens have illustrated the critical effect of papain, pepsin, alkaline protease and other commercial proteases for degrading several key protein allergens such as 7S protein α subunits and 11S soybean globulin [12]. However, these proteases showed low degradation efficiency due to their specificity against these protein allergens. The diverse functions of food enzymes support their uses in food processing; however, their insufficient stability, substrate specificity and activity can directly influence their involved applications, or result in unsatisfactory products after food processing.

## 3. Challenges for Food Enzyme Engineering

Naturally existing proteins usually show limitations during practical uses as a result of unsatisfactory activity or stability [58,59], which has motivated researchers to find solutions. During food processing, the conditions of high salt, high concentrations of organic solvents or high temperatures can be harsh for food enzymes. These adverse conditions mainly affect the half-life and activity of enzymes. Enzyme engineering aims to provide solutions for insufficient enzyme stability, unsatisfactory specific activity and weak solvent tolerance. Food enzyme dosages have strict standards set by organizations such as the EFSA and FDA [14]. Therefore, strengthening enzyme performance is necessary. In addition, modified enzymes with better environmental tolerance or activity can optimize the processing protocol, and reduce energy consumption by rapidly accomplishing the processing task or avoiding bacterial contamination.

Many studies have attempted to engineer food enzymes using direct evolution or rational design (Table 2) to overcome enzyme property limitations and expand their applications. The initial motivation for food enzyme modification was to meet specific applications, such as engineering the thermostability of microbial transglutaminase to satisfy its application for tofu and fish ball processing, since this processing involves a cool-down process prioritized to enzyme addition [27]. Meanwhile, previous reports suggested that protein thermostability is correlated with various harsh environmental tolerances [60]. Hence, improving enzyme thermostability may positively affect its overall stability under different conditions. More importantly, reinforcing an enzyme’s thermostability and activity can extend its potential utilities. For example, modification of substrate specificity can make microbial transglutaminase specifically label one site of human hormone, which supports its potential industrial application for polymer–drug conjugation [61]. The great effort being paid for food enzyme engineering has benefited their applications, such as in engineering the thermostability of phytase that improved its use as an animal feed supplement (usually pre-mixed at 75–95 °C) [55].

Whether using directed evolution or rational design for enzyme engineering, the resulting mutation sites are limited. Therefore, it is relatively difficult to obtain dramatic reinforcement of the target property using conventional engineering methods compared with screening enzymes from novel organisms, such as finding thermostable enzyme variants from thermophilic bacteria [71], or finding cold adaptive enzymes from deep-sea creatures [72]. In the past five years, achievements in using protein modeling and AI-based energy calculation for the rational design or de novo generation of functional proteins was conspicuous [22,33], especially for using de novo design to screen high-affinity protein binders, including drug proteins and antibodies [31,33]. De novo-designed proteins usually have high thermostability, since they always pursue the lowest energy structures. Meanwhile, de novo design-based novel enzyme screening dramatically reduces the difficulty of searching for potential functional enzymes from the gene library. However, de novo techniques assisting food enzymes engineering have been reported relatively less. De novo design using AI technology may provide more possibilities for expanding enzyme functions (including specific catalytic ability and stability) by generating non-existent proteins naturally. Moreover, recent progress for de novo design protein binders have highlighted the potential of directly generated enzymes with target functions.

## 4. De Novo Design Inspired by Highly Accurate Protein Modeling

Acknowledging the basis for protein modeling is the key to understanding protein de novo design. The protein evolution reinforced by AI has achieved great success, and led to the protein-based industry entering a new era [58,73]. Currently, AI-based tools have shown a significant impact on proteomics and structural biology, such as annotating proteins [2], searching functional proteins from the bacterial genome [74], predicting enzyme activity [75], modeling protein structures [22] and designing proteins [32]. Protein evolution reinforced by AI has contributed to the protein-based industry, which is entering a new era [58,73]. The pre-trained AI models used for predictions can save computational time and the required computing resources, which have benefited from their efficiency [76]. Machine learning (ML) and deep learning (DL) are critical subfields of AI, while the revolution of DL-based techniques dramatically benefited protein modeling and design in the past five years [22,32,77].

Most of the currently available protein design methods have adopted approaches used for protein modeling [21,26,78]. For instance, one of the examples involves treating the residue–residue contact map of the target protein as an image segmentation task, inspired using Convolutional Neural Network (CNN) [79] for protein folding prediction [80]. This method is commonly used for protein modeling and de novo design by extracting protein features [22,32,77]. Protein sequential arrangement or functional motif decomposition was thought to be like human language, which can be organized to represent certain meanings [81]. Accordingly, Natural Language Processing (NLP) methods originally used for human language translation were applied to resolving protein folding and de novo design tasks, by extracting features from protein sequences [22,82].

Accurate protein structural modeling is a complex task. A protein consists of amino acids which form its primary structure, and the residue–residue interactions within protein chains drive the formation of its secondary structure, including alpha helixes and beta-strands. A protein’s tertiary structure is ultimately formed due to the spatial arrangement of its secondary structures. Experimental techniques such as crystallography [83], nuclear magnetic resonance (NMR) [84] and Cryo-EM [85] have contributed to the Protein Data Bank (PDB) [86] dataset. However, due to the high cost of resolving protein structures, the currently available data comprise less than 200 thousand natural proteins, less than UniProt-deposited (more than 230,000 thousand) natural proteins [1]. In this context, the need for acquiring protein structures has promoted the progress of protein modeling methods.

### 4.1. Template-Based Protein Modeling

Protein modeling can be either template- or non-template-based. MSA is a well-employed template-based method for structural modeling [87,88]. The use of MSA aims to count the matches, mismatches and gaps of aligned protein sequences for uncovering the coevolutionary features of the target sequence [89] (Figure 2). For template-based protein modeling, the conserved areas revealed by MSA show that the target sequence is aligned to available structural data, and guides local folding to prioritize the whole structural folding. MSA was used very early by MODELLER [88] and SWISS-MODEL [90] to guide structural modeling, and has subsequently been adopted by AI-based tools. AI techniques show significant advantages in extracting protein features and performing predictions (Figure 3). Protein modeling protocols have been optimized using AI-based techniques, such as conducting MSA [91] and generating residue–residue interaction (RRI) [92] networks.

AlphaFold-2 (AF-2) highlights that the DL model can assist protein modeling as experimentally resolved, with an average error of 1 Å [22]. Protein modeling using MSA is a key to providing better accuracy [93]. Most AI-based protein modeling tools adopted MSA as an initial step, including AF-2 [22], RF [77,87], and I-TASSER. AF-2 learned features from the combination of MSA and RRI networks to guide the structure modeling. AF-2 developed a variant of Transformer [94] named Evoformer, which mainly uses “Attention” mechanism for feature extraction and progression. “Attention” mechanism was developed for language (sequence) translation (from encoder to decoder). It has been used to recognize conserved regions of the input sequence, and to guide the template selection in protein modeling [95]. On the other side, the features of RRI maps that were built depending on amino acids distribution and their steric interaction information (represented by a 2-dimensional distance map) were also extracted using “Attention” mechanism. Solely depending on RRI maps (trained networks), guided protein folding enabled AF-2 to perform non-template-based structure modeling. Generally, AF-2 adopts a combined search function that relies on MSA and RRI information to guide the template matches, followed by the “recycling step” that uses the output structure predicted by the network to match the structural labels, in order to guide model training. The protocol of AF-2 has inspired the invention of RF [77], which focuses on speeding up the prediction time by developing a concise MSA feature extraction step.

Another classic method, “multiple threading alignments” was adopted by I-TASSER [87], although I-TASSER adopted MSA as an initial modeling step. The “threading” method was used to evaluate the fitness of smaller sequential and structural fragments with the template protein backbone. The “threading”-based template search has also been widely used for protein de novo design for accommodating structural fragments, including the methods such as Rosetta match [26] and FixBB [96]. The key for template-based protein modeling is to find reliable local compartments, since fragment assembly is thought of as an early stage of whole protein folding [97].

### 4.2. Non-Template-Based Protein Modeling

In classic protein modeling methods, the non-template-based method was mainly stochastic sampling-based. Protein was thought to fold into its energy-minimized state in solution [98]. Monte Carlo (MC) and Markov Chain Monte Carlo (MCMC) simulations were commonly used to simulate atom movements and interactions, which were used for protein ab initio folding [99,100,101]. Currently, MC and MCMC methods are commonly used for whole structural optimization, rather than for performing protein folding or for modeling protein regions that are highly disordered [102]. The main drawback of non-template-based protein modeling is the computing speed, since both MC and MCMC use extensive computing resources [76]. In addition, ab initio folding cannot guarantee modeling accuracy either. Therefore, non-template-based protein modeling methods purely relied on MC, and MCMC was gradually abandoned.

For AI-based non-template-based protein modeling, RRI information used to train protein modeling networks brought out the possibility of training non-template-based models. RRI information used for training integrated the model, including AF-2 [22], RF [77], trRosetta [103] and I-TASSER [87]. A dramatic limitation of training networks solely depending on RRI information is the requirement of massive available structural data. However, currently available structural data may not be sufficient for training a solely RRI-based model, due to PDB having many reductant structures or proteins with high identities. Meanwhile, proteins such as membrane proteins only comprise a minor portion of the whole dataset, which can result in biased predictions after training [104]. Therefore, MSA-guided fragments assembly is still necessary for protein modeling, such as for membrane proteins, despite generating MSA in computational resources consumption.

Meta AI developed ESMFold without requiring the MSA step [105]. This model was trained with protein sequential and structural information using the “Attention” mechanism, and passes the input sequence directly to the DL block during prediction progression without an MSA session. This non-template search prediction has enabled fast computing of ESMFold, approximately 60 times faster than AF-2, despite its lower accuracy than AF-2 [105]. Until this review was organized, Meta AI announced that 617 million structures were modeled, and provided public access; meanwhile, the AF-2 database currently provides over 200 million structures. High modeling accuracy can be supportive for subsequent protein design. Meanwhile, building a DL model for protein modeling has inspired the construction of AI-based protein de novo design protocols such as RFdesign [31] and Hallucination [32]. We collectively listed AI-based protein modeling methods by describing their architecture and utilities in Table 3.

## 5. De Novo Design of Food Enzymes

Protein de novo design refers to the use of rational ways to design novel proteins that do not exist naturally. It is described as solving the inverse problem of protein modeling [110]. De novo design can be used to design sequences compatible with the given structural backbone [111,112], generate a missing block within a given structure [30,31], or “hallucinate” de novo structures with sequences [32] (Figure 4). Previously, physics-based modules such as Rosetta Match [26], Fixbb [96] and Remodel [113] (Table 4) have been successfully implemented to search optimized sequences based on a given structure (Table 4). These physics-based methods adopt structural fragments replacement or insertion. They can also be used for large secondary structure replacement, using Rosetta-based modules such as FunFolDes [114] and LUCS [115]. Physics-based methods have mainly been applied in the past ten years. Users can design blueprints and assemble protein by fragment accommodation attempts using TopoBuilder [34]. However, physics-based de novo design methods require large sampling sizes, which is time-consuming. AI-based methods emerged to enable fast computing, highly accurate structural modeling, and the design of functional enzymes. The successful implementation of AF-2 brought out possibilities for DL to resolve de novo design tasks. Currently, AI-methods such as Recurrent Neural Network (RNN) [116], CNN, Graph Neural Networks (GNN) [117] and Generative Adversarial Nets (GAN) [118] have highly participated in integrating de novo design models (Table 4). Using AI-based techniques for de novo design functional proteins is showing an apparent upward trend [119]. In the following section, we review the progression of de novo designs for food enzyme engineering, and introduce AI-based de novo design methods available for further uses.

### 5.1. Current Solutions

Generating functional food enzymes using de novo design has been successfully implemented. Note that de novo design food enzymes can be traced to 20–30 years ago, such as de novo designs of superoxide dismutase-like enzyme [123] and esterase [124]. The initial challenge for enzyme de novo design is to sustain its activity, whereas earlier algorithms showed that accommodating catalytic residues in the pocket was difficult and uncontrollable [123,124]. As a result, the functions of de novo-designed food enzymes were not satisfactory. Novel algorithms were brought out to overcome this limitation, by focusing on the accommodation of enzyme active sites. The standout algorithm was Rosetta Match, which takes advantage of the structural fragments dataset and fragments’ replacement strategy for designing protein functional motifs [78,125]. This algorithm has been used for the de novo design of pullulanase [126] and esterase [127]. It should be noted that the combined use of several techniques such as consensus design, MSA and fragments’ replacement contributed to the high success rate for designing novel enzymes. Several reports showed that the designed food enzymes notably outperformed native enzymes in terms of their catalytic activity [127] or stability [126]. However, it was reported that de novo designs of food enzymes were mostly physics-based, which highly relies on computational resources and empirical factors.

AI-based de novo design techniques merged to address these limitations and enable fast design. It was recently reported that ProteinGAN [35] adopted GAN to generate de novo protein sequences. ProteinGAN was trained using protein sequence data through the “one-hot” method, in order to convert these sequences into a digital array. The network was built using ResNet [128] (CNN derived) and “Attention” mechanism for extracting features from input sequences (by discriminator), and generating novel sequences (by generator). In the case study, the sequences from the malate dehydrogenase family were used, and the authors showed that 24% of de novo-designed sequences were expressed experimentally with enzymatic functions. The initial successes of AI-based techniques for enzyme de novo design successes have triggered the rapid development of algorithms. However, many novel AI-based de novo design techniques have not been applied to food enzymes, which still show great potential.

### 5.2. AI-Based De Novo Design Techniques

#### 5.2.1. Protein Hallucination

Protein “Hallucination” [32] was inspired by Google DeepDream, and was introduced in 2020. Hallucination can be used to generate de novo protein structures with compatible sequences, without requesting the protein backbone. Hallucination generates a stabilized structure using a random input sequence, and it adopts the method from trRosetta to describe the 2-dimensional RRI map of the input sequence [103]. An independent test used MCMC for tracking RRI map changes upon mutations, and revealed a sharpened RRI map indicating a stabilized structure. Based on the results, Hallucination employed multiple rounds of iterations by introducing mutations and tracking the RRI map changes to find the optimized structure with a compatible sequence. The prediction efficiency of Hallucination was experimentally characterized by circular dichroism (CD) spectroscopy; there were 62 out of 129 samples solubly expressed that showed the desired 2-dimensional structures as predicted. The expressed proteins showed high thermostability, with an average melting temperature above 70 °C, and the crystallized structures highly assembled the predictions. The released code enables users to input a sequence and adjust the length of the output sequence. However, the limitation is that the user cannot control the motif insertion position, since the output sequence was randomly mutated based on the input sequence.

#### 5.2.2. RFDesign

The accuracy of structural modeling was significantly improved using AF-2 and RF compared with trRosetta. RFDesign took advantage of AF-2 and RF [31], which provided an updated version of protein Hallucination, as well as an “inpainting” module for rebuilding missing sequences and structures by enabling reserved functional regions. The authors attempted to use RF [77] and AF-2 [22] for generating protein features rather than trRosetta [103], and it showed significant advantages. RFDesign hallucination adopted a similar approach as the previous version, to reward those mutations that could stabilize the structure. For the inpainting design module, an iteration method mimics the “recycled” step of AF-2, and RF (RF-Nov05-2021 version) was brought out to continuously introduce mutations to the output structure, followed by overall folding refinement and simultaneously scoring the output structure.

Previously, de novo design proteins were always based on input proteins that could not design proteins with specific functions. To resolve this challenge, RFDesign hallucination [31] developed a combined training loss that scored the repulsive and binding forces between the input protein and its binding partner (can be protein or ligand); those mutations that showed correct binding behavior were retained for iteration. In the case study, the authors used RFDesign hallucination to successfully design the interactive surface of programmed cell death protein 1 (PD-1), and reinforced its binding affinity against programmed cell death ligand 1 (PD-L1). In the released package, users can design specific sites within the input sequence and indicate the receptor file. The protein Hallucination family solved two major challenges in protein de novo design, including generating de novo motifs and designing functional protein binders.

#### 5.2.3. ProteinMPNN

ProteinMPNN [33] was used for the de novo design of compatible sequences based on the input structure backbone. ProteinMPNN was built using a modified Transformer [94] network, which adopted the encoder block to extract features, and the decoder block to translate these features to “readable information”. ProteinMPNN was trained using protein backbone information, including C-alpha atom–atom distance, orientation and backbone dihedral angles, rather than requesting MSA information. The training method contributed to the high capacity for modeling single sequences, despite the fact that AF-2 [22] and RF [77] highly required MSA information for protein folding prediction. ProteinMPNN showed a sequence recovery rate of 52.4%, and could serve for the the design of protein monomers or cyclic oligomers. In a recent report, ProteinMPNN was successfully used for designing binder protein, highlighting its great potential [129].

#### 5.2.4. DenseCPD

DenseCPD [120] is a sequence design method based on the input structure. This model is trained using protein atomic information. The training structures are prepared by splitting the protein box into small voxels containing only 1 atom (1 Å). DenseCPD learns the atom distribution information from the structures using DenseNet [130] (CNN derived), and predicts the probability of amino acids that build the input protein backbone. This approach displayed higher accuracy than the later released ABACUS-R [131], despite ABACUS-R relying on Transformer to extract more information from both protein sequence and structure. The aim of using DenseCPD is to find the most suitable sequences for the protein backbone, and this model is currently supported only for tasks submitted online.

#### 5.2.5. Unsupervised Learning Methods

The large sample size of deposited protein structural and sequential data promoted the development of unsupervised learning (UL) [132] methods. ProtGPT2 [121] is a pre-trained language-based UL model which generates protein sequences based on training sequences. ProtGPT2 employs a modified Transformer by taking advantage of only the decoder side. This network was trained using UniRef50 [133], which contains approximately 49 million protein sequences that are highly diverse. Compared with ProteinGAN [35], the trained ProtGPT2 model is able to generate de novo sequences within a few seconds based on the user input sequences, which is convenient.

Diffusion model (DM) [134] was recently brought out as a generative network, through adding noises and iterative denoising to recover the targets. DM previously showed state-of-art performance for synthesizing images and generating videos. Namrata and Tudor implemented DM to generate de novo protein design, which aims to build missing areas or find compatible sequences [122]. This model was trained and adopted protein information, including protein full-atom coordinates, protein sequences and amino acid side-chain conformations, and showed reliable performance during the validation tests.

## 6. Limitations of De Novo Design Techniques and Opportunities for Food Enzyme Engineering

There are still apparent challenges for the generation of functional enzymes using de novo design. Successful implementation of de novo design was achieved with high-affinity protein binders, such as IL-2/IL-15 homologues [135] and ACE2 protein homologues, within 67 days [136]. This evidence showed that the means for designing desired proteins was partially addressed. It is easy to de novo generate thermostable variants of target enzymes, since AI models were trained to output the lowest energy structures [32]. However, enzyme-catalyzed reactions require a certain distance between the catalytic residue of an enzyme, and the residue being catalyzed of a substrate. These distance constraints may not prioritize generating novel enzymes, although we can still use an additional protocol to filter binding complexes that satisfy the distance. For protein binder design, the binding or repulsive forces can be used to indicate the protein–partner binding affinity [137]. However, in enzyme-triggered catalysis, the relationship between enzyme–substrate binding affinity (*k*_m_) and catalytic activity (*k*_cat_) are still under debate. Such consequences highlight the fact that much knowledge remains unknown about enzyme-induced catalysis.

The success rate is also a problem for de novo design. The success rate of designing malate dehydrogenase using ProteinGAN only achieved 24% [35]. The DL-models, such as ProteinGAN and DLKcat [75], have mainly been used for extracting information from protein sequences, which means the enzyme structures are not correlated with prediction; however, a lack of structural indication may negatively impact prediction accuracy. Note that DLKcat [75] was specifically used for predicting enzyme *k*_cat_ (turnover number of an enzyme), which can predict *k*_cat_ changes toward specific substrates while enzyme sequences change. Nonetheless, whether using physics-based or AI-based methods, de novo-designed proteins remained with low expression rates, or were difficult to fold [31,138]. Meanwhile, few studies crystallized their predicted structures, which showed noticeable RMSD variations [32,139]. To those who are designing binding partner tasks, minor RMSD differences between designed and actual structures can result in non-binding or bad contacts (such as strong repulsive forces). De novo designing enzymes with desired functions is a huge challenge.

Challenges always coexist with opportunities. De novo design have shown great potential for generating non-naturally existing and diverse functional variants, which provides possibilities for their currently involved and novel applications. A few factors would benefit DL-model architecture for generating various functional food enzymes: firstly, training the model by adding structural information; secondly, collecting more information related to enzymes and their *k*_cat_ against certain substrates; and thirdly, investigating correlations between complex binding and enzyme activity.

## 7. Conclusions

This review described the functions and applications of food enzymes, and introduced the need for engineering enzymes to satisfy their applications or expand their utility horizon. The limitations of using conventional enzyme engineering methods are evident, as fewer mutation rates may make it difficult to strengthen the target functions of enzymes. The advances in AI-based protein modeling and de novo design methods were reviewed. The successful implementations of de novo design for functional protein binders highlight the possibilities of using de novo design for functional enzymes. The challenges for the de novo design of enzymes come from the limited knowledge of enzyme–substrate binding behaviors and their correlated activities; the architecture of the models; and insufficient training data. These issues await future investigation.

## Figures and Tables

**Figure 1 ijms-24-03827-f001:**
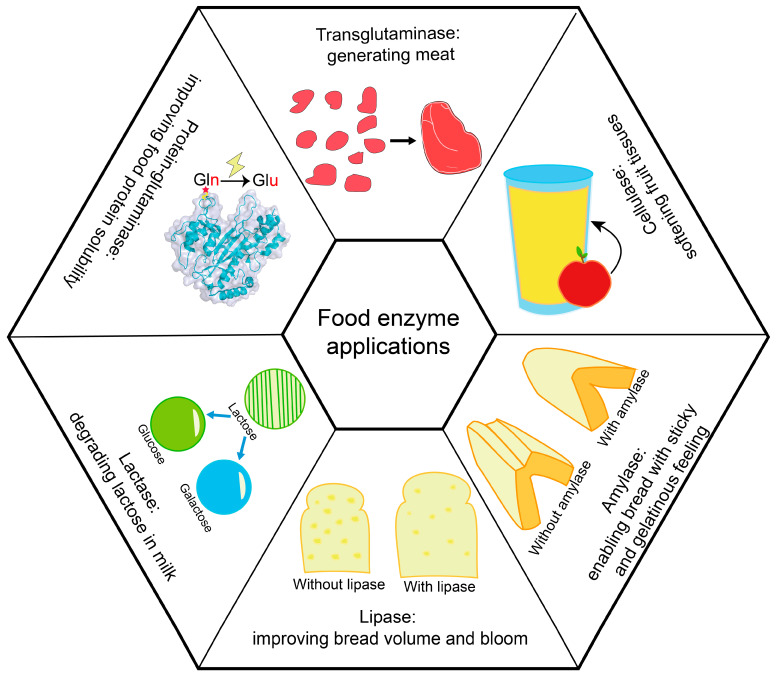
Sample applications of food enzymes. This figure lists six commonly used food enzymes. Transglutaminase (structure obtained from Protein Data Bank, ID: 1IU4, generated by PyMol) catalyzes the formation of heteropeptide bonds between the γ-amide group of the glutamine residue in the protein and the ε-amino group of Lys. Cellulase hydrolyzes cellulose to produce glucose and oligomeric fiber. Amylase hydrolyzes the glycosidic bonds inside starch. Lipase hydrolyzes triglycerides to glycerol and fatty acids. Lactase catalyzes the hydrolysis of β-D-galactoside and α-L-arabinoside. Protein-glutaminase catalyzes the deamidation of Glu residues in proteins. Protein structure generated using PyMol (Schrödinger, New York, NY, USA).

**Figure 2 ijms-24-03827-f002:**
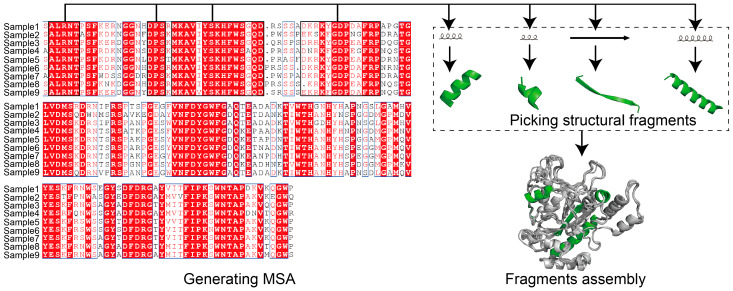
Using MSA for structural modeling. Multiple sequences alignment (MSA) describes aligning protein sequences to highlight the conserved region. For protein modeling, MSA information is used to guide searching of structural fragment templates prior to whole structure folding. Protein sequence alignment was generated using Jalview (Andrew Waterhouse and Geoff Barton’s group, Dundee, Scotland). Protein structure generated using PyMol (Schrödinger, New York, NY, USA).

**Figure 3 ijms-24-03827-f003:**
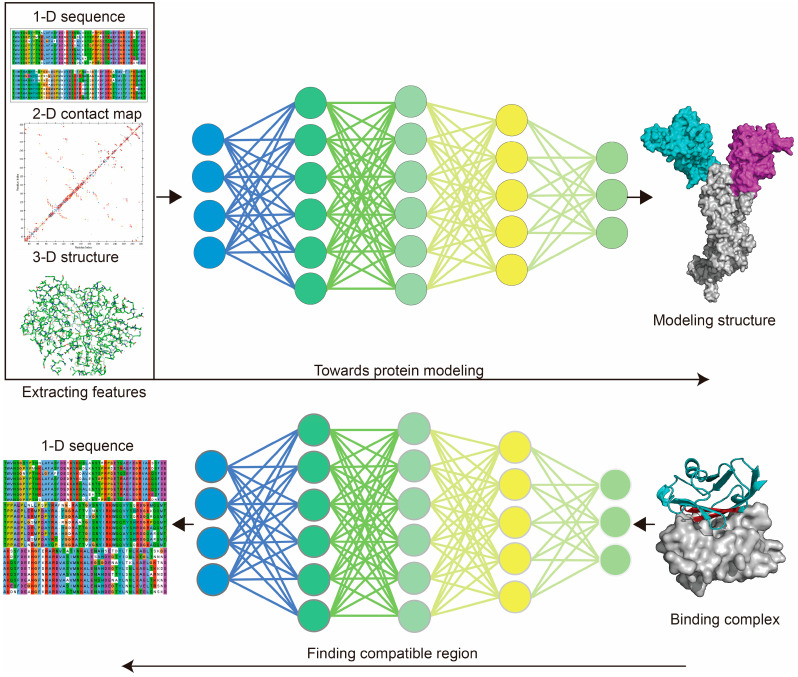
Deep learning used for structural modeling and design. For protein modeling, deep learning (DL) is used to learn sequence information (or MSA features) from the 1-dimensional sequences, residue–residue interaction information from the 2-dimensional contact map, or 3-dimensional structural information based on input structures, through aligning with the target structure (DL labels) to train the model. Protein de novo design takes advantage of protein modeling models by finding the compatible structure (and sequences) of the target block within a given protein to design novel proteins. Protein sequence alignment generated using Jalview. Protein 2-dimensional contact map generated using Discovery Studio 2019. Protein structure generated using PyMol.

**Figure 4 ijms-24-03827-f004:**
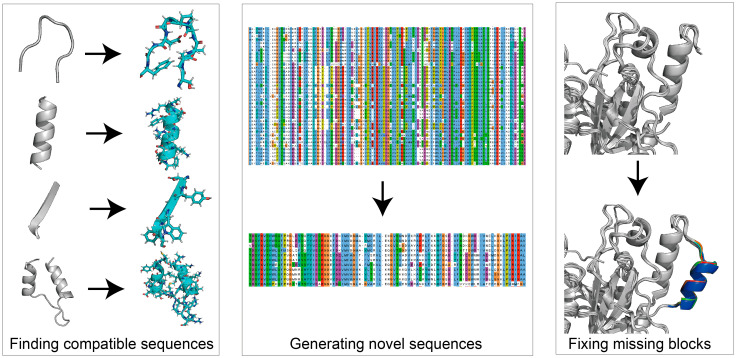
Utilities of protein *de novo* design. Protein de novo design can be used for finding compatible sequences of target areas (ProteinMPNN), generating novel sequences based on input sequences (ProteinGAN), or fixing missing blocks of given structures (Hallucination). Protein sequence alignment was generated using Jalview. Protein structure generated using PyMol.

**Table 1 ijms-24-03827-t001:** Typical functions of food enzymes.

Enzyme (EC Number)	Catalytic Reaction	Commercial Source
Transglutaminase (EC 2.3.2.13)	Catalyzing the formation of heteropeptide bonds between the γ-amide group of the glutamine residue in the protein and the ε-amino group of Lys [36].	*Streptomyces mobaraense*
Laccase (EC 1.10.3.2)	Catalyzing single-electron oxidation of phenols, aromatic amines, and other electron-rich substrates [45].	*Aspergillus oryzae, Mycyceliophora thermophila*
Protein-glutaminase (EC 3.5.1.44)	Catalyzing the deamidation of Glu residues of proteins [46].	*Chryseobacterium proteolyticum*
α-Amylase (EC 3.2.1.1)	Hydrolyzing α-1,4-glycosidic bonds inside starch [38].	*Bacillus licheniformis, Thermus Aquaticus*
Lactase (EC 3.2.1.108)	Catalyzing the hydrolysis of β-D-galactoside and α-L-arabinoside [40].	*Bacillus subtilis, Bifidobacterium bifidum*
α-Glucanase (EC 3.2.1.59); β-Glucanase (EC 3.2.1.73)	Hydrolyzing α/β-glucans [47].	*Bacillus subtilis, Bacillus amyloliquefaciens*
Phosphatidylinositol-specific phospholipase C (EC 3.1.4.11)	Hydrolyzing the phosphodiester bond of phosphatidylinositol to diacylglycerol and water-soluble phosphoinositol [48].	*Pseudomonas fluorescens*
Polygalacturonase (EC 3.2.1.15)	Catalyzing pectin molecule α-(1,4)-polygalacturonic acid cleavage [49].	*Trichoderma reesei, Aspergillus tubingensis*
Pectinesterase (EC 3.1.1.11)	Hydrolyzing pectin to produce pectinic acid and methanol [43].	*Trichoderma reesei, Aspergillus tubingensis*
endo-β-1,4-xylanase (EC 3.2.1.8)	Hydrolyzing xylan molecule β-1,4-glycosidic bonds [50].	*Trichoderma reesei, Thermopolyspora flexuosa*
Lipases (EC 3.1.1.3)	Hydrolyzing triglycerides to glycerol and fatty acids [41].	*Trichoderma reesei, Fusarium oxysporum*
4-α-glucanotransferase (EC 2.4.1.25)	Catalyzing the breaking of α-1,4-glycosidic bonds and the transfer of α-glucan residues within or between molecules [51].	*Aeribacillus pallidus*
Rennin (EC 3.4.4.3)	Hydrolyzing the peptide bond between Phe105-Met106 of κ-casein in milk [52].	*Kluyveromyces lactis*
Cellulase (EC 3.2.1.4)	Hydrolyzing cellulose to produce glucose and oligomeric fiber [37].	*Trichoderma reesei*
Glucose isomerase (EC 5.3.1.18)	Catalyzing isomerization of glucose to fructose [53].	*Streptomyces, Bacillus subtilis*
α-glucosidase (EC 3.2.1.20)	Hydrolyzing the glycosidic bond of the non-reducing end of polysaccharides or converting the α-1,4-glycosidic bond of oligosaccharides into α-1,6-glycosidic bonds [53].	*Saccharomycetes, Aspergilusniger*
Glucose oxidase (EC 1.1.3.4)	Oxidizing β-D-glucose to become gluconic acid and hydrogen peroxide [44].	*Aspergillus niger*
Subtilisin (EC 3.4.21.62)	Hydrolyzing proteins into amino acids [54].	*Bacillus subtilis*
Phytase (EC 3.1.3.8)	Catalyzing the removal of phosphate groups by inositol hexaphosphate [55].	*Natuphos*

**Table 2 ijms-24-03827-t002:** Enzyme modification and the modification aims.

Enzyme and Source	Effect of Best Variant	Aims and Reference
Transglutaminase (*Streptomyces mobaraenesis*)	*T*_m_ and specific activity increased by 3.4 °C and 67.8%.	Processing tofu and fish balls at high temperatures. [27]
Glucoamylase (*Talaromyces leycettanus*)	*T*_m_ and specific activity increased by 9 °C and 305.4%.	Inducing the conversion of starch to glucose at high temperatures. [62]
Alpha-amylase (*Rhizopus oryzae*)	*t*_1/2_ (55 °C) increased by 2.55-fold.	Optimizing winemaking protocol. [63]
Cellulase (*Penicillium canescens*)	*t*_1/2_ (60 °C) increased by 3.4-fold.	Catalyzing the formation of gentiooligsaccharide at high temperatures. [64]
Serine protease (*Pseudomonas aeruginosa*)	*T*_m_ and specific activity increased by 5 °C and 1.4-fold.	Protease treatment at high temperatures enables fast processing and avoids bacterial contamination. [65]
Lipase (*Yarrowia lipolytica*)	*t*_1/2_ (50 °C) increased by 70%.	Optimizing grain and oil-processing protocol. [66]
Endoglucanase (*Bipolaris sorokiniana*)	Specific activity increased by 1.5-fold.	Enabling rapid food processing. [67]
Phytase (*Escherichia coli*)	Residual activity improved by 78.4% at 90 °C.	Used as animal feed supplement. [68]
Glucose isomerase (*Thermoanaerobacter ethanolicus*)	Specific activity increased by 2-fold.	High-fructose corn syrup one-step biosynthesis. [69]
β-glucanases (*Bacillus terquilensis*)	Improved acidic tolerance, and increased specific activity by 45%.	Serving food mashing process. [70]

*T*_m_: Melting temperature, a temperature point at which protein undergoes a reversible folding or unfolding transition. *t*_1/2_: Half-life (*t*_1/2_) for proteins at a specific temperature.

**Table 3 ijms-24-03827-t003:** AI-based protein modeling tools.

Name	Description	Ref
AlphaFold-2	Accurate, structures can be directly downloaded from a public dataset. Slow for protein modeling using source code. Database accessed from: https://alphafold.com/ (accessed on 5 February 2020) https://www.uniprot.org/ (accessed on 5 February 2020)	[22]
ESMFold	Accurate, structures can be directly downloaded from a public dataset. Database accessed from: https://esmatlas.com/about#download_dataset (accessed on 5 February 2020)	[106]
RoseTTAFold	Accurate, support for uploading up to 20 sequences for modeling. Relatively fast for protein modeling using source code. Webserver: https://robetta.bakerlab.org/ (accessed on 5 February 2020)	[77]
I-TASSER	Accurate, support for online uploading modeling tasks and using source code. Webserver: https://zhanggroup.org/I-TASSER/ (accessed on 5 February 2020)	[107]
trRosetta	Accurate, support for online uploading modeling tasks and using source code. Webserver: https://yanglab.nankai.edu.cn/trRosetta/help/ (accessed on 5 February 2020)	[103]
A-Prot	Only support for source code modeling. Source code: https://github.com/arontier/A_Prot_Paper (accessed on 5 February 2020)	[108]
Colossal-AI	Only support for source code modeling. Source code: https://github.com/hpcaitech/ColossalAI (accessed on 5 February 2020)	[109]

**Table 4 ijms-24-03827-t004:** De novo design methods.

Name	Description	Ref
Match	Physics-based, structural-based, aims at designing de novo functional enzymes using fragment attempts.	[26]
Fixbb	Physics-based, structural-based, fit for short area design.	[96]
Remodel	Physics-based, structural-based, fit for short area design, can be used for protein reassembling.	[113]
FunFolDes	Physics-based, structural-based, blueprint-based whole protein de novo design.	[114]
LUCS	Physics-based, structural-based, fit for designing loop-helix-loop, loop-strand-loop.	[115]
TopoBuilder	Physics-based, structural-based, blueprint-based whole protein de novo design.	[34]
Protein Hallucination	AI-based, de novo design of whole protein structures with compatible sequences based on input sequence.	[32]
RFDesign	AI-based, “inpainting” module: designing short blocks based on given structure; “hallucination” module: designing short blocks based on a given structure, can be used for designing functional motifs, supporting receptor and donor structure.	[31]
ProteinMPNN	AI-based, fast designing compatible sequences using an input structure.	[33]
DenseCPD	AI-based, only supports uploaded tasks online, online server: http://protein.org.cn/densecpd.html	[120]
ProteinGAN	AI-based, GAN model for real-time generating sequences based on a set of input protein sequences (within the same protein family).	[35]
ProtGPT2	AI-based, pre-trained model for generating sequences based on input sequence.	[121]
Diffusion model-based	AI-based, pre-trained model generating protein structures.	[122]

## Data Availability

All data related to this study have been included in the manuscript.

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
