# Peer review of "Possibilities of Using De Novo Design for Generating Diverse Functional Food Enzymes"

_ijms, 2023, doi:10.3390/ijms24043827_

Round 1

Reviewer 1 Report

The authors reviewed the food enzyme applications, food enzyme engineering basics with an emphasis on using de novo design for food protein modelling, along with the current limitations.

The work has been carried out with care, it is novel, original and backed-up with relevant references. Since the topic is interesting and rapidly growing, I believe that it could be an interesting contribution to this scientific field after thorough revision.

However, the English level is unsatisfactory and masks all the advantages and the seriousness of the paper. I recommend detailed revision of English by a professional. Specific comments are skipped, since there are too many mistakes. Wrong use of suffixes (-ing, -ed), tenses, wrong word order in the sentence, unclear sentences like: food enzymes are widely used in the processing of various food processing, and also unclear paragraphs, wrong use of propositions, adverbs, adjectives (also in figure 1 – not with lipase instead of without lipase), use of words like welly instead of well,  degreding instead of degrading, etc. Also, not all abbreviations mentioned in the text are explained, ...

Author Response

Dear Reviewer, 

Thank you for providing comments concerning our manuscript.

We have carefully read all the comments, and the manuscript has been revised as suggested. The answer to each comment is in the attached file. We hope that the revised version will be acceptable for publication.

All the best

Reviewer 2 Report

Dear authors,

I consider your paper "Possibilities of using de novo design for generating diverse functional food enzymes" is not appropriate, in the present form, for being published in IJMS.

Below you can find some of my observations/suggestions:

1. English language requires some changes, it is not always easy to understand the phrases (for example: "………………are natural biocatalysts participated in biological reactions" / "Natural existed proteins usually showed limitations during practical uses result from unsatisfied activity or stability……………..")

2. Significance of Tm (firstly appeared in in Table) and t1/2 (half-time)

3. Figures from the article – where are they from? There are realized by the authors or there are taken from other articles? In this case, reference has to be given, and/or the permission for being used, obtained from the journal

4. What information were obtained using de novo design for the enzymes you proposed in the title? I think you have to give more information on these enzymes; in the review I found only generalities. I imagined, starting from the title of your article, that you will present the problems solved, using this method, for the functional food enzymes... 

Author Response

Dear Reviewer,

Thanks for your comments concerning our manuscript.

We have carefully read all the comments, and the manuscript has been revised as suggested. The answer to each comment is in the attached file. We hope that the revised version will be acceptable for publication.

All the best

Round 2

Author Response

Dear reviewer,

Thank you very much for carefully reading our manuscript and providing comments, we really appreciate it. The manuscript has been revised as suggested, and we hope this version could be satisfactory.

All the best

Reviewer 2 Report

Dear Authors,

I saw the revised version of your paper, and now I think it is appropriate for being published.

Success with your future work!

Author Response

Dear Reviewer,

Thank you for your comment and encouragement.

All the best